# MSH6/2 and PD-L1 Expressions Are Associated with Tumor Growth and Invasiveness in Silent Pituitary Adenoma Subtypes

**DOI:** 10.3390/ijms21082831

**Published:** 2020-04-18

**Authors:** Shinsuke Uraki, Hiroyuki Ariyasu, Asako Doi, Ken Takeshima, Shuhei Morita, Hidefumi Inaba, Hiroto Furuta, Noriaki Fukuhara, Naoko Inoshita, Hiroshi Nishioka, Naoyuki Nakao, Shozo Yamada, Takashi Akamizu

**Affiliations:** 1First Department of Internal Medicine, Wakayama Medical University, Wakayama 641-8509, Japan; shinnojo0723@gmail.com (S.U.); minami@wakayama-med.ac.jp (A.D.); kt10948@wakayama-med.ac.jp (K.T.); smorita@wakayama-med.ac.jp (S.M.); inaba@wakayama-med.ac.jp (H.I.); hfuruta@wakayama-med.ac.jp (H.F.); akamizu@wakayama-med.ac.jp (T.A.); 2Department of Hypothalamic and Pituitary Surgery, Toranomon Hospital, Tokyo 105-8470, Japan; n-fukuhara@hotmail.co.jp (N.F.); nishioka@tokyo-med.ac.jp (H.N.); 3Department of Pathology, Tokyo Metropolitan Geriatric Medical Center, Tokyo 173-0015, Japan; inonao3939@gmail.com; 4Department of Neurological Surgery, Wakayama Medical University, Wakayama 641-8509, Japan; nnakao@wakayama-med.ac.jp; 5Hypothalamic and Pituitary Center, Moriyama Neurological Center Hospital, Tokyo 134-0088, Japan; syamadays11@hotmail.com; 6Kuma Hospital, Center for Excellence in Thyroid Care, Kobe, 650-0011, Japan

**Keywords:** pituitary adenoma, mismatch repair gene, MSH6, MSH2, tumor immunity, PD-L1

## Abstract

Mismatch repair genes *mutS homologs 6/2* (*MSH6/2*) expressions are involved in tumor growth and *programmed cell death 1 ligand 1* (*PD-L1*) expression in tumor immunity, but the direct association with pituitary adenomas (PAs) is not well understood. We aimed to clarify the effects of MSH6/2 and PD-L1 expression on tumor proliferation and invasiveness in nonfunctioning (NF) PAs. We performed immunohistochemistry to classify the NFPAs into gonadotroph adenoma (GAs), silent corticotroph adenomas (SCAs), null cell adenoma (NCAs), and pituitary transcription factor 1 (PIT1) lineage PAs. We evaluated *MSH6/2* and *PD-L1* mRNA expressions in NFPAs by real-time PCR (*n* = 73), and statistically analyzed the expressions and clinicopathological factors. We also investigated the effect of MSH6 knockout on PD-L1 expression in AtT-20ins and GH3. MSH6/2 expressions were significantly lower in invasive NFPAs than in non-invasive NFPAs, and lower in SCAs and NCAs than in GAs. *MSH6/2* expressions were positively associated with *PD-L1* expression. *PD-L1* expression was significantly lower in invasive NFPAs than in non-invasive NFPAs, and lower in SCAs and NCAs than in GAs. Although *MSH6/2* expressions also tended to be lower in PIT1 lineage PAs than in GAs, PIT1 lineage PAs expressed *PD-L1* equivalently to GA, which was unlike SCAs and NCAs. MSH6 knockout in AtT-20ins and GH3 significantly decreased *PD-L1* expression (75% and 34% reduction, respectively) with cell proliferation promotion. In conclusion, differences in *MSH6/2* and *PD-L1* expressions of SCAs, NCAs, and PIT1-lineage PAs from those of GAs appear to contribute to their clinically aggressive characteristics, such as more proliferation and invasiveness.

## 1. Introduction

Pituitary adenomas (PAs) are common intracranial tumors comprising 15% to 20% of all brain tumors, which have been recently referred to as “pituitary neuroendocrine tumors” (Pit-NETs) [1,2]. PAs are predominantly benign tumors, but they include highly proliferative and invasive tumors, such as aggressive PAs and pituitary carcinomas [3,4]. Regarding proliferation, when non-functioning pituitary carcinomas (NFPAs) are pathologically classified based on 2017 World Health Organization (WHO) classifications, null cell adenoma (NCA), silent corticotroph adenoma (SCA), and PIT1-lineage PA are reportedly highly proliferative and invasive [5,6,7,8,9]. Genetic abnormalities (*aryl hydrocarbon receptor-interacting protein* (*AIP*) gene, *ubiquitin specific protease 8* (*USP8*) gene mutation) have been reported to be related to tumor development and proliferation in a proportion of functioning PAs (acromegaly, Cushing’s disease) [10], but the mechanism of pathogenesis and growth in most PAs is unclear. Pharmacotherapy for PAs is therefore currently limited, especially for nonfunctioning pituitary adenomas (NFPAs). We recently reported that reduced expression of mismatch repair (MMR) genes *MSH6*/*MSH2* directly promotes pituitary tumor proliferation via the ataxia telangiectasia and Rad3-related-checkpoint kinase 1 (ATR-Chk1) pathway [11,12]. In PAs, MMR genes may be target molecules for future drug therapy.

On the other hand, MMR gene expression has been suggested to be associated with prediction of the effects of immune checkpoint inhibitors (ICIs). For example, in various cancers, such as colorectal cancer, ICIs such as pembrolizumab and nivolumab have been reported as being effective against tumors lacking the MMR protein encoded by the MMR gene [13,14,15]. PD-L1 expression in tumors is also an important factor in predicting the effects of ICIs. ICIs are described as being highly effective for non-small cell lung carcinoma, renal cancer, and bladder cancer with high expression of PD-L1 [16,17,18]. Regarding pituitary tumors, in one case report, the ICIs ipilimumab and nivolumab were effective against a pituitary carcinoma with negative expression of MSH6 [19]. However, the relationship between MMR gene expression and PD-L1 expression in PAs has not been fully elucidated.

Based on these findings, we hypothesize that MSH6/2 and PD-L1 expressions could affect pituitary tumor proliferation, and invasion by pathological classification of NFPAs and ICIs may be effective for NFPA subtypes with decreased MSH6/2 expression and increased PD-L1 expression. In this study, for the first time, we analyzed the correlation between *MSH6/2* and *PD-L1* mRNA expression levels and clinical and pathological factors related to tumor proliferation and invasiveness using tissue from human NFPAs.

## 2. Results

### 2.1. Expressions of MSH6/2 and PD-L1 mRNA Are Significantly Lower in Invasive NFPAs Than in Non-Invasive NFPAs

In the present study, NFPAs with Knosp grades 1–2 (*n* = 37) were considered to be non-invasive tumors and NFPAs with Knosp grades 3–4 (*n* = 36) were considered to be invasive tumors, in accordance with a previous report [20]. We compared the expression levels of *MSH6/2* and *PD-L1* in the invasive and non-invasive NFPAs. The expressions of *MSH6/2* and *PD-L1* were significantly lower in invasive NFPAs than in non-invasive NFPAs (respectively, *p* < 0.001, *p* < 0.001, and *p* < 0.05; Figure 1A).

To examine the association between *MSH6/2* and *PD-L1* mRNA expressions and MIB-1-labeling index (LI), an indicator of pituitary tumor growth, we performed simple linear regression analysis and compared the expression levels of each gene in the MIB-1 < 3% group and ≥ 3% group. There was no significant correlation between *MSH6/2* expressions of MIB-1 LI, as in our previous report (*n* = 73; Appendix A) [12]. On the other hand, *PD-L1* mRNA expression was positively correlated with MIB-1 LI, and tended to be higher in the MIB-1 ≥ 3% group than in MIB-1 < 3% group (*n* = 73; Appendix A).

### 2.2. Expressions of MSH6/2 and PD-L1 mRNA Are Significantly Lower in Silent Corticotroph Adenomas and Null Cell Adenomas Than in Gonadotroph Adenomas

To compare expressions of *MSH6/2* and *PD-L1* mRNA by histological subtype of NFPAs, we divided 73 NFPAs into GA (SF1 lineage), SCA (TPIT lineage), NCA, and others (PIT1 lineage) including silent somatotroph adenoma, silent lactotroph adenoma, silent thyrotroph adenoma, and PIT1-positive plurihormonal adenoma, according to the 2017 WHO classification [21,22,23]. *MSH6/2* mRNA expressions were significantly lower in SCAs and NCAs than in GAs (*p* < 0.01 and *p* < 0.05, respectively; Figure 1B). Meanwhile, although the expression of *MSH6/2* also tended to be lower in others (PIT1 lineage) than in GAs, we found no statistically significant difference in *MSH6/2* expressions between GA and others (PIT1 lineage).

*PD-L1* mRNA expression was significantly lower in SCAs and NCAs than in GAs (*p* < 0.01 and *p* < 0.05, respectively; Figure 1B). Meanwhile, we found no statistically significant difference in *PD-L1* expression between GA and others (PIT1 lineage). Although the expression of *MSH6/2* mRNA in others (PIT1 lineage PAs) tended to be lower than in GA, the expression of *PD-L1* mRNA in others was equivalent to in GA, which was unlike SCAs and NCAs (Figure 1B). Fifteen NFPA samples obtained at Wakayama Medical Hospital were immunohistologically analyzed for PD-L1 protein expression. This revealed that the expression of PD-L1 protein in each NFPA subtype was GAs (*n* = 4), 13.8% (± 15.4); SCAs (*n* = 4), <1%; NCAs (*n* = 2), <1%; PIT1-lineage PAs (*n* = 5), 18.6% (± 27.3), respectively, indicating PD-L1 mRNA study results. Representative images are shown in Figure 1C.

Additionally, the expressions of *MSH6/2* and *PD-L1* mRNA were lower in NFPA subtypes with Knosp grade 3–4 than in NFPA subtypes with Knosp grade 1–2, although a significant *p* value could not be obtained due to the small number of samples in each subtype (Appendix A).

### 2.3. Expression of MSH6 and MSH2 mRNA Is Positively Associated with Expression of PD-L1 mRNA in Clinically Nonfunctioning Pituitary Adenomas

To investigate how expressions of *MSH6* and *MSH2* are involved in *PD-L1* expression in clinically NFPAs, we performed simple linear regression analysis on the relationship between *MSH6/2* expression and *PD-L1* expression (*n* = 73). Expressions of *MSH6* and *MSH2* were positively associated with expression of *PD-L1* (respectively, *R* = 0.65, *p* < 0.0001 and *R* = 0.66, *p* < 0.0001; Figure 2). In addition, this positive correlation was also observed by pathological classification of NFPAs (Appendix A).

### 2.4. Knockout of MSH6 Expression by Cells by Clustered Regularly Interspaced Short Palindromic Repeats/Cas9 (CRISPR-Cas9) System Reduces PD-L1 mRNA Expression with Promotion of Cell Proliferation in AtT-20ins and GH3 Cells

To determine whether *MSH6* expression affects pituitary tumor growth and *PD-L1* expression, we investigated the effect of gene ablation of *MSH6* on *PD-L1* expression and cell proliferation in AtT-20ins and GH3 cells.

We generated MSH6-knockout AtT-20ins and GH3 cells using the Cells by Clustered Regularly Interspaced Short Palindromic Repeats/Cas9 (CRISPR-Cas9) system, and confirmed that MSH6 was knocked out in AtT-20ins and GH3 by western blotting (Figure 3). While gene disruption of *MSH6* resulted in 75% decrease in PD-L1 mRNA expression in AtT-20ins cells, corticotroph cell line, they resulted in only 34% decrease in GH3 cells, somatotroph cell-line (Figure 3). Regarding cell proliferation, this gene disruption led to 39% increase in AtT-20ins cell number and 77% increase in GH3 cell number (Figure 3). As in our previous reports [12], gene ablation of MSH6 reduced the expression of ATR, a cell cycle moderator (data not shown). We therefore propose that increase in cell number of AtT-20ins and GH3 cells was promoted via the ATR-chk1 pathway.

## 3. Discussion

In general, MMR genes play crucial roles in the correction of mismatch errors resulting from nucleotide misincorporation by DNA polymerase. A reduction in MMR gene expression can increase the risk of tumor genesis by microsatellite instability and higher mutational load [24]. In pituitary tumors, according to recent exome sequencing studies, only a small number of somatic point mutations are observed, and MSI is not detected [25]. We reported that reduced expression of MMR genes *MSH6/MSH2* directly promotes pituitary tumor proliferation via the cell cycle regulatory mechanism ATR-Chk1 pathway, not via the typical mechanism reported above [12]. Regarding MMR genes and pituitary tumors, the *MSH6* gene was reportedly hypermethylated in pituitary macroadenomas, and a corticotroph pituitary carcinoma developed in a patient with Lynch syndrome. High prevalence of invasive PAs was also shown in patients with Lynch syndrome in a nation-wide survey based in Sweden [26,27].

The concept of aggressiveness in PAs is a composite of invasion, proliferation, regrowth, and recurrence. Although clinically NFPAs are generally less proliferative and invasive, some NFPAs are relatively proliferative and invasive, and effective drug therapies for them are limited. In this study, we therefore used patient samples of clinical NFPAs, specifically to pathologically silent pituitary adenomas. We considered NFPAs with Knosp grade 1–2 as non-invasive tumors and NFPAs with Knosp grade 3–4 as invasive tumors, in accordance with the previous report [20], and compared the expression levels of *MSH6/2* mRNA between the two groups. The expressions of *MSH6/2* were significantly reduced in invasive PAs compared with non-invasive PAs. These results suggest that decreased expressions of *MSH6/2* were partially involved in the molecular mechanism of invasive PAs. On the other hand, recent reports showed that since the introduction of endoscopy in transsphenoidal surgery, surgeons can distinguish between invasive and non-invasive by directly visualizing the medial wall of the cavernous sinus (MWCS) [28]. It is therefore important to take a visual inspection into account, but a limitation of this study is that we could not consider it in this study because of the lack of available data in our database. Further study is needed in the future.

According to 2017 WHO classification, NFPAs are clinically divided into GA, SCA, NCA, and PIT1-lineage PA; reported frequencies were 73%, 16%, 1%, and 9%, respectively [6,9]. By histological subtype, compared to GA, SCA, NCA, and PIT1-lineage PA have more aggressive behavior such as giant adenoma, marked cavernous sinus invasion, and greater rate of recurrence [6,9]. In the current study, the expression of *MSH6/2* in SCAs and NCAs was significantly lower than that in GAs, which may be part of the molecular mechanism causing proliferative and invasive features. Meanwhile, there was also no significant correlation between *MSH6/2* expression and MIB-LI. These results suggest that the reduction in *MSH6/2* expression may cause a decrease in apoptosis rather than cell cycle promotion in tumor growth through the cell cycle regulatory mechanism ATR-Chk-1 pathway, because MIB-1 LI is an indicator of tumor cell division [12]. MIB-1 LI may not accurately represent pituitary tumor proliferation because the value is calculated from the highest percentage of immunoreactive tumor cells in several evaluated areas of a pituitary tumor.

MMR gene expressions were reported to be involved in predicting the effects of ICIs for malignant tumors [13,14,15]. PD-L1 expression in tumors is also reported to be an important factor in predicting the effectiveness of ICIs [16,17,18]. In malignant tumors such as breast carcinomas, endometrial carcinomas, and colorectal cancers, PD-L1 expression was increased in cases of microsatellite instability showing MMR protein deficiency or abnormality of MMR mechanism [29,30,31]. We therefore expected similar results in NFPAs. Contrary to our expectation, expressions of *MSH6* and *MSH2* mRNA were positively associated with expression of *PD-L1* mRNA in clinical NFPAs, and the same correlations were also observed by pathological subtype. Unfortunately, in this study, we could not reveal the mechanism in detail, but we have some suggestions. Pituitary tumors could have their own unique mechanism that controls their expressions, which is unlike other malignant tumors. Alternatively, pituitary tumors are generally classified as being benign, including the series in the current study, so there may be a regulatory mechanism in *MSH6/2* and *PD-L1* mRNA expressions in benign tumors as NFPAs that differs from that in malignant tumors. No reports have yet examined the correlation between *PD-L1* and MMR gene expressions in other benign tumors, so further studies are needed, especially regarding the association of their expressions in PAs as benign tumors and in pituitary carcinomas as malignant tumors.

Regarding the relationship between *PD-L1* and clinicopathological factors in NFPAs, invasive PAs also had significantly lower PD-L1 expression than non-invasive PAs. PD-L1 expressed in tumors has been shown to bind to PD-1 expressed in cytotoxic T cells (CTLs), which suppresses CTL activity and causes tumor immune tolerance [32]. In other words, in NFPAs, tumor proliferation is promoted by the reduction of the expression of *MSH6/2*. *PD-L1* expression also decreases, suggesting that tumor immunity may be relatively functional in NFPAs. This differs from malignant tumors such as breast cancer, and it may contribute to the majority of pituitary tumors being benign [29,30,31]. By subtypes of NFPAs, SCAs and NCAs showed a significant decrease in *PD-L1* expression compared to GAs, suggesting that the tumor immunity in SCAs and NCAs may be relatively functional compared to GAs. In general, however, it is suggested that SCAs and NCAs have more aggressive behavior compared to GAs [6,9]. Factors other than tumor immunity, such as the reduction of *MSH6/2* expression and cell cycle regulators, could play a more important role in the proliferation and invasiveness in SCAs and NCAs. On the other hand, PIT1-lineage PAs, unlike SCAs and NCAs, have relatively high *PD-L1* expression levels. This could be because the proliferative and invasive behaviors of PIT1-lineage PAs might involve two factors: tumor proliferation promotion due to decreased *MSH6/2* expression, and tumor immunity tolerance due to relatively high PD-L1 expression. The variation in *PD-L1* expression in NFPA subtypes may create differences in tumor immunity among NFPA subtypes. Meanwhile, there was a significant positive correlation between *PD-L1* expression and MIB-1 LI. This result remains controversial; it has been discussed both positively and negatively [33,34]. In one report with a negative result, intra-tumoral PD-1 expression positively correlated with MIB-1 LI. Aggressive PAs were confirmed in a recent study to downregulate antigen-presenting genes and suppress immune responses [35]. It was thought, therefore, that aggressive PA with high MIB-1 LI is more immune-related, and may be treated with immunotherapy such as PD-1/PD-L1 blockade.

Finally, based on the results of human NFPAs, in vitro studies with AtT-20ins, mouse corticotroph tumor cell line, and GH3, rat somatotroph tumor cell line were performed. In MSH6 knockout AtT-20ins, MSH6 knockout promoted cell proliferation with decreased expression of MSH2 and ATR mRNA, as we previously reported [12]. This result was also observed in GH3. In addition, MSH6 knockout significantly decreased *PD-L1* mRNA expression in AtT20ins and GH3. However, the decrease in PD-L1 mRNA expression by MSH6 knockout was greater in AtT20ins than in GH3. This result demonstrated the relationship between *MSH6* and *PD-L1* mRNA expression in human NFPAs, especially SCA and PIT1-lineage PAs. To date, two different mechanisms of PD-L1 regulation on tumor cells have been reported: innate immune resistance and adaptive immune resistance [36]. Innate immune resistance means that constitutive oncogenic signaling within tumor cells such as ALK-rearranged and EGFR-mutant lung cancer results in the upregulation of PD-L1 expression [37,38]. In contrast, adaptive immune resistance means that local inflammatory signals such as interferons-ɤ and cytokines produced by active anti-tumor immune response induces PD-L1 expression on tumor cells [39,40]. In molecular mechanisms such as innate immune resistance, the results of this in vitro investigation suggests that PD-L1 may be downregulated by decreased MSH6 expression in pituitary tumors, which could lead to a tumor microenvironment that facilitates tumor immunity in pituitary tumors.

In conclusion, *MSH6/2* expressions of SCAs, NCAs, and PIT1-lineage PAs compared to GAs are thought to contribute to clinically invasive and proliferative characteristics. The molecular mechanism of the difference in clinical features of NFPAs was partially elucidated. Notably, there is some possibility that decreased expressions of *MSH6/2* might be useful for predicting the proliferation and invasiveness of NFPAs. The difference in PD-L1 expression in NFPA subtypes could be involved in tumor growth and invasion via tumor immunity tolerance. In this study, we could not provide sufficient evidence that ICIs would be useful in treating PAs. However, there is a case with pituitary carcinoma in which the treatment with ICIs was effective despite the decreased expression of PD-L1, similarly to our results [19]. This study is limited in that the human NFPAs used in this research were benign and the samples did not include pituitary carcinoma. The accumulation of real-world evidence is needed to evaluate the effectiveness of ICIs against Pas, including aggressive PAs and pituitary carcinomas. Further studies on the detailed mechanisms of MSH6/2 expressions and *PD-L1* expression in PAs are also warranted.

## 4. Materials and Methods

### 4.1. Ethics Statement

Informed consent was obtained from all patients for the collection, analysis, and publication of personal and clinical data, and for the use of pituitary tumor specimens in gene expression and immunohistochemical studies. This study was performed with approval of the ethics committees of Wakayama Medical University and Toranomon Hospital (approval numbers 133 and 1804 (project identification code), 15 January 2016).

### 4.2. Patient Samples

Of the patients with clinical NFPAs treated between February 2016 and February 2018 at Wakayama Medical University Hospital or Toranomon Hospital, 89 patients consented to our study. Of these, 16 patients requiring re-operation were excluded, so 73 patients were included in our study. Patient clinical, imagological, and histopathological characteristics are shown in Table 1 and Appendix A.

### 4.3. Immunohistochemistry (IHC)

Formalin-fixed paraffin-embedded (FFPE) tissue sections of pituitary tumors were histopathologically and immunohistochemically examined. We used the following antibodies: growth hormone (GH, 54/9 2A2; BioGenex, Fremont, CA, USA), prolactin (polyclonal; DAKO, Santa Clara, CA, USA), beta-thyroid-stimulating hormone (beta-TSH, 0042; DAKO), adrenocorticotropic hormone (ACTH, 02A3; DAKO), beta-follicle-stimulating hormone (beta-FSH, C10; BioGenex), beta-luteinizing hormone (beta-LH, C93; DAKO), Ki-67 (MIB-1,790-4286; Ventana, Tucson, AZ, USA), cytokeratin (CAM5.2; BECTON, Franklin Lakes, NJ, USA), and PD-L1(22C3 pharmDx; DAKO). Immunohistochemistry was performed using the BenchMark Ultra automated system (Ventana, Oro Valley, AZ, USA) as previously described [11]. Several pathologists reviewed specimens and classified pituitary adenomas as suggested in the 2017 WHO classification [21,22,23]. Pituitary adenomas were considered to be hormone-negative if <1% of the cells stained positively for any adenohypophyseal hormones [6]. Tumors originally classified as hormone-negative NFPAs and a few silent adenomas were subjected to a further panel of antibodies against adenohypophyseal cell lineage transcription factors including steroidogenic factor-1 (SF1, N1665; Perseus Proteomics, Tokyo, Japan), pituitary transcription factor 1 (PIT1, sc-136034; Santa Cruz, Dallas, TX, USA), and t-box pituitary transcription factor (TPIT, HPA072686; Atlas Antibodies AB). PD-L1 expression was calculated as the ratio of positive cells to all cells among pituitary tumor cells.

### 4.4. Measurement of MSH6/2 Gene and PD-L1 Gene mRNA Expression

Total RNA was extracted using the TRIzol reagent. cDNA was synthesized using the TaqMan™ Reverse Transcription Reagents (Applied Biosystems, Foster City, CA, USA). Quantitative RT-PCR (qRT-PCR) analyses were carried out to detect mRNA levels (*MSH2*, *MSH6*, and *PD-L1*) using SYBR Green Real-Time PCR Master Mixes (Applied Biosystems). *GAPDH* was used as an internal control. Primers for qRT-PCR are shown in Appendix A.

### 4.5. Generation of MSH6 Knockout AtT-20ins and GH3 Cells by Clustered Regularly Interspaced Short Palindromic Repeats/Cas9-Mediated Genome Editing Technique

The open-access software programs NCBI Gene (https://www.ncbi.nlm.nih.gov/gene) and CRISPR direct (https://crispr.dbcls.jp), were used to design sgRNAs targeted to *MSH6* exon 1 (NC_000083.6). The sgRNA targeting MSH6 (5′-TCAGCCTTCATCATGTCCCG-3′ or 5′-GGCACTCGGCGACACCAAG-3′) was cloned into pGuide-it-ZsGreen1 vector (no 632601; Clontech Laboratories, Inc., CA, USA). The plasmid vector (pGuide-it-ZsGreen-sgMSH6) was sequenced to confirm successful ligation. AtT-20ins and GH3 was transiently transfected with pGuide-it-ZsGreen-sgMSH6 using Lipofectamine 3000 (Life Technologies, Carlsbad, CA, USA). Enhanced green fluorescent protein (EGFP)-positive cells were sorted with FACS Aria II (BD Biosciences, Franklin Lakes, NJ, USA) and clones were generated as single colonies by limiting dilutions. All isolated colonies were evaluated for the expression of MSH6 using Western blotting. We carried out Western blotting as previously described [41]. Briefly, for protein analysis, cells were lysed in M-PER buffer (Thermo Scientific, Rockford, IL, USA) mixed with a protease inhibitor. The lysis solution was mixed and centrifuged at 15,000× *g* for 5 min at 4 °C. The supernatant was collected and quantified by colorimetric protein assay (BioRad, Berkeley, CA, USA). Western blots were performed using 4–12% Bis-Tris (NuPage) precast gels on Invitrogen XCell SureLock Mini-Cell modules. Blocking, incubation with antibodies and washing were carried out in Tris-buffered saline with 0.05% (*v*/*v*) Tween-20 and 5% (*w*/*v*) non-fat dry milk. Primary antibodies against target proteins were diluted in the blocking solution (anti MSH6 antibody (M3646; DAKO); 1:2000, anti β-Actin (Sigma-Aldrich, St. Louis, MO, USA; 1:5000). Secondary antibodies were diluted in a blocking solution to detect the species-specific portion of the primary antibodies. Antibody binding was detected using LAS-3000 mini (FUJI Film, Tokyo, Japan), and quantified by densitometry using ImageJ (NIH, Bethesda, MD, USA). β-Actin was used as the loading control. All experiments with recombinant DNA technology were approved by the DNA Recombination Experiment Committee of Wakayama Medical University (approval number 30–81 (project identification code), 15 April 2019).

### 4.6. Cell Culture and Treatment Schedule

Our cell culture and treatment schedule followed the same procedures as in our previous study [12]. Briefly, after wild-type AtT-20ins and GH3 and MSH6 KO AtT-20ins and GH3 were incubated for 48 h in 24 well or 96 well plates under standard conditions (37 °C, 5% CO_2_), total RNA was isolated from each cell and cDNA was synthesized. Quantitative RT-PCR (qRT-PCR) analyses to detect *PD-L1* mRNA levels were carried out using SYBR Green Real-Time PCR Master Mixes (Applied Biosystems), and *GAPDH* was used as an internal control. Primers for qRT-PCR are shown in Appendix A. Cell proliferation assays were performed in 96 well plates using the Cell Counting Kit-8 (CCK-8; Dojindo, Rockville, MD, USA).

### 4.7. Statistical Analysis

Data were statistically analyzed using simple linear regression analysis (Kendall rank correlation coefficients), unpaired *t* test, and one-way analysis of variance followed by Dunnett’s test for multiple comparisons. Values of *p* < 0.05 were considered statistically significant. Analyses were performed using a commercially available statistical package (JMP Pro version 13.1, SAS Institute, Cary, NC, USA).

## Figures and Tables

**Figure 1 ijms-21-02831-f001:**
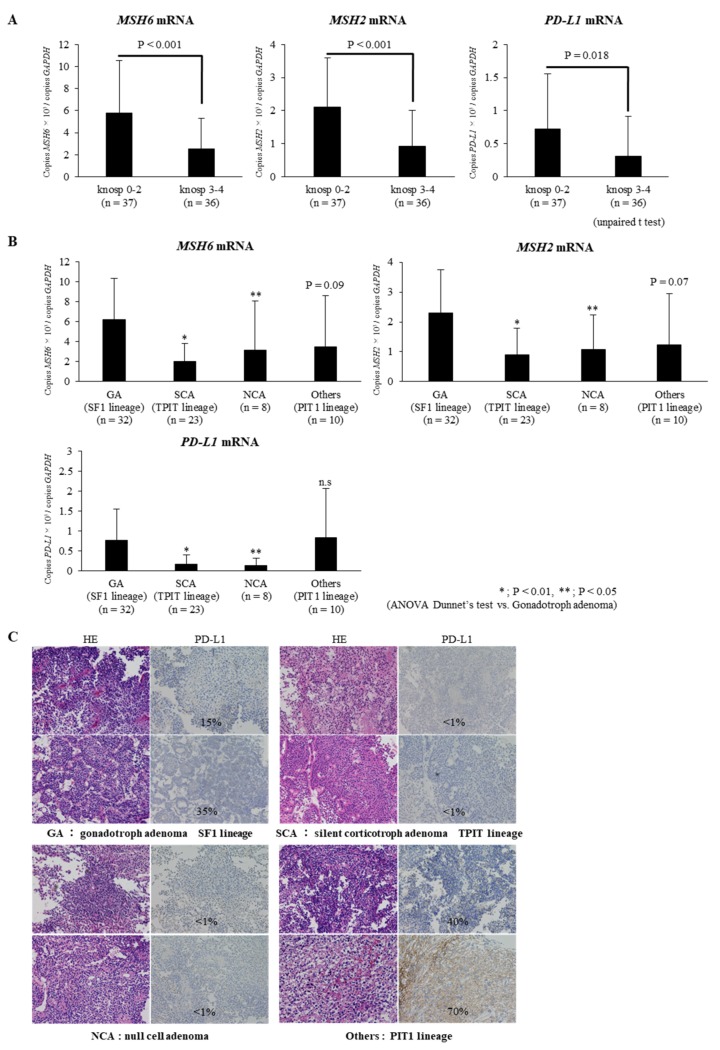
Relationship between *MSH6*, *MSH2*, and *PD-L1* expressions and clinicopathological factors in clinically nonfunctioning pituitary adenomas. (**A**) *MSH6*, *MSH2*, and *PD-L1* mRNA expression between Knosp grade 0–2 and Knosp grade 3–4 in clinically nonfunctioning pituitary adenomas. The expression of *MSH6/2* and *PD-L1* mRNA was significantly lower in NFPAs with Knosp grade 3–4 than in NFPAs with Knosp grade 1–2 (*p* < 0.001, *p* < 0.001, and *p* < 0.05, respectively). (**B**) *MSH6*, *MSH2*, and mRNA expression between histological subtypes in clinically nonfunctioning pituitary adenomas. The expression of *MSH6/2* and *PD-L1* mRNA was significantly lower in SCAs and NCAs than in GAs (*p* < 0.01 and *p* < 0.05, respectively). Although there was no statistically significant difference in the *MSH6/2* and *PD-L1* mRNA expression between GA and others (PIT1 lineage), the expression of *MSH6/2* mRNA in others tended to be lower than in GA. On the other hand, the expression of *PD-L1* mRNA in others was equivalent to that in GA, which was unlike SCAs and NCAs. * *p* < 0.01, ** *p* < 0.05. (**C**) PD-L1 protein expression between histological subtypes in clinically nonfunctioning pituitary adenomas. In representative samples, the relationship with *PD-L1* expression between histological subtypes was also observed in the immunohistochemical analysis.

**Figure 2 ijms-21-02831-f002:**
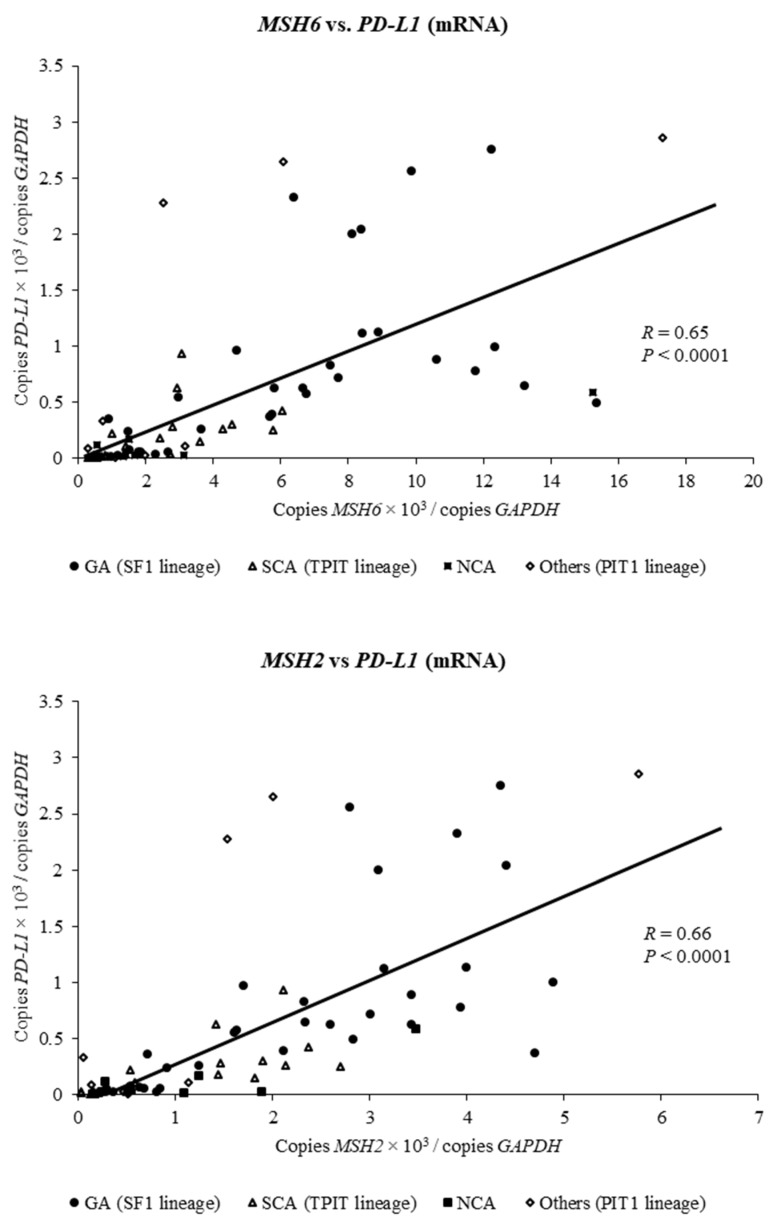
Correlation between *MSH6* and *MSH2* mRNA expression and *PD-L1* mRNA expression in clinically nonfunctioning pituitary adenomas. The relationship between *MSH6* and *MSH2* expressions and *PD-L1* expression in NFPAs was examined using simple linear regression analysis. *MSH6* and *MSH2* expression was positively associated with *PD-L1* expression (*R* = 0.65, *p* < 0.0001 and *R* = 0.66, *p* < 0.0001, respectively).

**Figure 3 ijms-21-02831-f003:**
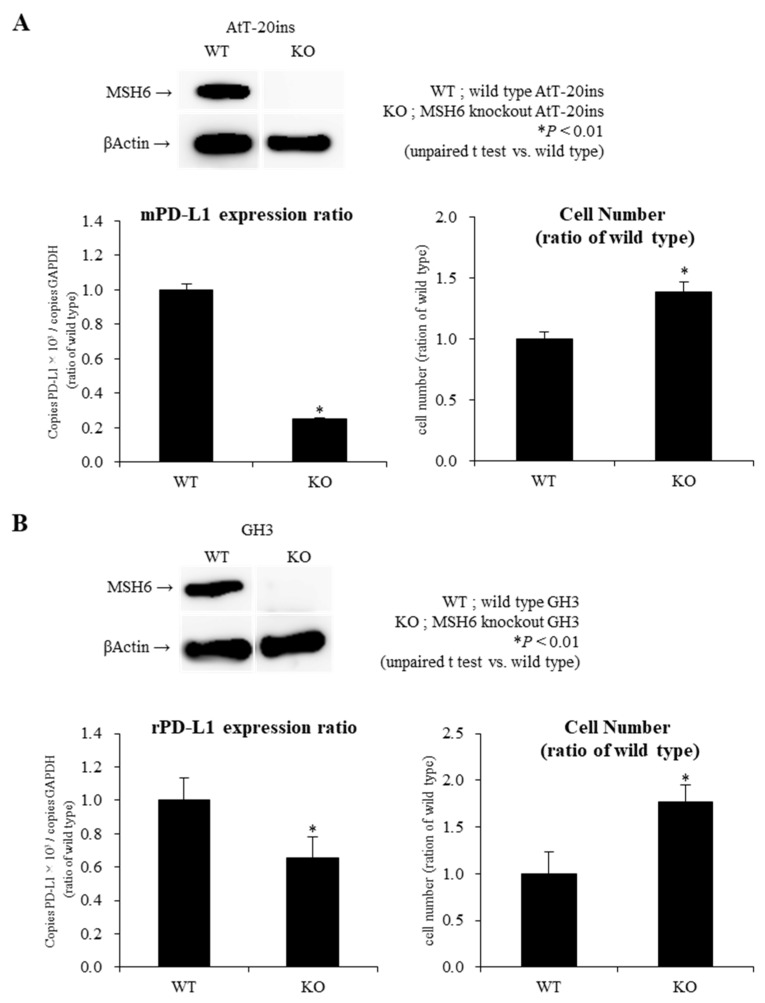
Knockout of MSH6 expression by Cells by Clustered Regularly Interspaced Short Palindromic Repeats/Cas9 (CRISPR-Cas9) system reduces *PD-L1* mRNA expression with promotion of cell proliferation in AtT-20ins (**A**) and GH3 (**B**) cells. We generated MSH6-knockout AtT-20ins and GH3 cells using the CRISPR-Cas9 system. MSH6 was knocked out in AtT-20ins and GH3 by Western blotting. The knockout of MSH6 significantly reduced PD-L1 mRNA expression with promotion of cell proliferation in AtT-20ins and GH3 cells by qRT-PCR and Cell Counting Kit-8 analysis. * *p* < 0.01.

**Table 1 ijms-21-02831-t001:** Clinical, radiological, and histopathological characteristics of 73 patients with clinically nonfunctioning pituitary adenoma.

**Pathological Phenotype**	**Number**
Gonadotroph adenoma	32
Silent corticotroph adenoma	23
Silent thyrotroph adenoma	2
Silent lactotroph adenoma	4
Silent plurihormonal (PRL, GH) adenoma	4
Null cell adenoma	8
Total	73
**Clinical, Radiological, and Histopathological Data**	
Male/Female	36/37
Age	52.2 ± 15.7 years *
Micro/Macro	0/73
MIB-1 LI (%)	1.6% ** (0.1–4.1)
	<3%: 65, ≧3%: 8
Knosp grade	grade0: 3, grade1: 19, grade2: 15, grade3: 22, grade4: 14

* mean ± SD ** median.

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
