# Peer review of "MSH6/2 and PD-L1 Expressions Are Associated with Tumor Growth and Invasiveness in Silent Pituitary Adenoma Subtypes"

_ijms, 2020, doi:10.3390/ijms21082831_

Round 1
Reviewer 1 Report
I have no futher commnets.
Author Response
Prof. Dr. Micheline Misrahi
Editor-in-Chief, "Molecular Endocrinology and Metabolism" Section,
Prof. Dr. Salvatore M. Corsello
Dr. Rosa Maria Paragliola
Guest Editors, Special Issue "Molecular Biology of the Pituitary",
The International Journal of Molecular Sciences
April 14, 2020
Dear Prof. Dr. Misrahi, Prof. Dr. Corsello and Dr. Paragliola,
  Thank you very much for your e-mail dated April 10, 2020, regarding our manuscript for publication as a Clinical Research Article in The International Journal of Molecular Sciences: “MSH6/2 and PD-L1 expressions are associated with tumor growth and invasiveness in silent pituitary adenoma subtypes (Manuscript ID: ijms-776467)”

  We appreciate the opportunity to resubmit our manuscript and we thank you and the reviewers for your thoughtful comments and suggestions. Based on this valuable feedback, we have carefully re-examined and revised our manuscript. Please find attached the revised version of our manuscript, along with our Response to Reviewers.
  All critical changes made to the text during this revision process have been highlighted in red, so that you and the reviewers may easily identify them. There are also some small changes throughout the manuscript that improve syntax, grammar and overall clarity. We have revised the formatting and form of our manuscript based on your directions.
  We hope that our manuscript is now suitable for publication as a Clinical Research Article in The International Journal of Molecular Sciences. We look forward to hearing from you at your convenience.
Sincerely,
Hiroyuki Ariyasu, M.D., Ph.D.
The First Department of Medicine,
Wakayama Medical University
811-1 Kimiidera, Wakayama, Japan 641-8509
Phone: +81-73-441-0625
E-mail: ariyasu@wakayama-med.ac.jp
Response to Reviewer #1:
We thank the Reviewer very much for their valuable comments and suggestions.

Reviewer 2 Report
The paper has significantly amended. Nevertheless, the writing should be checked again, especially the Results section in order to eliminate duplicates sentences, and the discussion remains somewhat confusing. As I previously suggested, table 1 would be more comprehensible in one column than in two.
Finally, the most important commentary is the contradiction between the lower PD-L1 expression in invasive and aggressive variants of pituitary tumors (sCT and NC tumors) compared with non-invasive and less aggressive PT (GTs) and the sentence in the discussion “SCAs and NCAs also showed a significant decrease in PD-L1 expression compared to GAs, suggesting that the tumor immunity in SCAs and NCAs may be relatively functional” and the sentence of the conclusions “Considering MSH6/2 and PD-L1 expression patterns in NFPAs, compared to GA, ICIs could be effective against SCA, NCA and PIT1 lineage Pas”.
Founded on their results, I believe that the document does not provide sufficient evidence to indicate treatment with ICIs in pituitary tumors, at least in non-aggressive tumors. Accordingly, the conclusions should be reexamined.
Author Response
Prof. Dr. Micheline Misrahi
Editor-in-Chief, "Molecular Endocrinology and Metabolism" Section,
Prof. Dr. Salvatore M. Corsello
Dr. Rosa Maria Paragliola
Guest Editors, Special Issue "Molecular Biology of the Pituitary",
The International Journal of Molecular Sciences
April 14, 2020
Dear Prof. Dr. Misrahi, Prof. Dr. Corsello and Dr. Paragliola,
Thank you very much for your e-mail dated April 10, 2020, regarding our manuscript for publication as a Clinical Research Article in The International Journal of Molecular Sciences: “MSH6/2 and PD-L1 expressions are associated with tumor growth and invasiveness in silent pituitary adenoma subtypes (Manuscript ID: ijms-776467)”
We appreciate the opportunity to resubmit our manuscript and we thank you and the reviewers for your thoughtful comments and suggestions. Based on this valuable feedback, we have carefully re-examined and revised our manuscript. Please find attached the revised version of our manuscript, along with our Response to Reviewers.
All critical changes made to the text during this revision process have been highlighted in red, so that you and the reviewers may easily identify them. There are also some small changes throughout the manuscript that improve syntax, grammar and overall clarity. We have revised the formatting and form of our manuscript based on your directions.
We hope that our manuscript is now suitable for publication as a Clinical Research Article in The International Journal of Molecular Sciences. We look forward to hearing from you at your convenience.
Sincerely,
Hiroyuki Ariyasu, M.D., Ph.D.
The First Department of Medicine,
Wakayama Medical University
811-1 Kimiidera, Wakayama, Japan 641-8509
Phone: +81-73-441-0625
E-mail: ariyasu@wakayama-med.ac.jp
Response to Reviewer #2:
We are grateful to the Reviewer for their valuable comments and suggestions. In response, we have revised the manuscript as follows:
Minor Comments
Reviewer’s Comment:
The paper has significantly amended. Nevertheless, the writing should be checked again, especially the Results section in order to eliminate duplicates sentences, and the discussion remains somewhat confusing. As I previously suggested, table 1 would be more comprehensible in one column than in two.
Our response:
 As the Reviewer indicated, we revised the Result section in order to eliminate duplicates sentences. In addition, we modified Table1 from two column to one.
Reviewer’s Comment:
Finally, the most important commentary is the contradiction between the lower PD-L1 expression in invasive and aggressive variants of pituitary tumors (sCT and NC tumors) compared with non-invasive and less aggressive PT (GTs) and the sentence in the discussion “SCAs and NCAs also showed a significant decrease in PD-L1 expression compared to GAs, suggesting that the tumor immunity in SCAs and NCAs may be relatively functional” and the sentence of the conclusions “Considering MSH6/2 and PD-L1 expression patterns in NFPAs, compared to GA, ICIs could be effective against SCA, NCA and PIT1 lineage Pas”.
Founded on their results, I believe that the document does not provide sufficient evidence to indicate treatment with ICIs in pituitary tumors, at least in non-aggressive tumors. Accordingly, the conclusions should be reexamined.
Our response:
 As the Reviewer indicated, we agreed to your suggestion.
We have therefore added to and modified the following sentences in the Discussion section:
(page 8, lines 221–232)
“By subtypes of NFPAs, SCAs and NCAs showed a significant decrease in PD-L1 expression compared to GAs, suggesting that the tumor immunity in SCAs and NCAs may be relatively functional compared to GAs. In general, however, it is suggested that SCAs and NCAs have more aggressive behavior compared to GAs [6, 9]. Factors other than tumor immunity, such as the reduction of MSH6/2 expression and cell cycle regulators, could play a more important role in the proliferation and invasiveness in SCAs and NCAs. On the other hand, PIT1 lineage PAs, unlike SCAs and NCAs, have relatively high PD-L1 expression level. This could be because the proliferative and invasive behaviors of PIT1 lineage PAs might involve two factors: tumor proliferation promotion due to decreased MSH6/2 expressions, and tumor immunity tolerance due to relatively high PD-L1 expression. The variation in PD-L1 expression in NFPA subtypes may give difference in tumor immunity among NFPA subtypes.”
(page 8-9, lines 261–268)
“In this study, we could not provide sufficient evidence that ICIs would be useful in treating PAs. However, there is a case with pituitary carcinoma in which the treatment of ICIs was effective despite the decreased expression of PD-L1, similar to our results [19]. This study is limited in that the human NFPAs used in this research are benign and the samples do not include pituitary carcinoma. The accumulation of real-world evidence is needed to evaluate the effectiveness of ICIs against PAs including aggressive PAs and pituitary carcinomas. Further studies on the detailed mechanisms of MSH6/2 expressions and PD-L1 expression in PAs are also warranted.”

This manuscript is a resubmission of an earlier submission. The following is a list of the peer review reports and author responses from that submission.
Round 1
Reviewer 1 Report
The authors submitted for publication an interesting and up to date paper focusing on new possible targets of immunotherapy of pituitary tumors by studying the expression of MSH 6/2 and PD-L1 in an acceptable series of invasive and non-invasive non-functioning tumors. In addition to compare the levels of expression between invasive and non-invasive tumors, they study the effect of KO of MSH6 in pituitary cell lines to evaluate its effect on PD-L1 expression, which strengthens their hypothesis that immune checkpoint inhibitors could be a promising therapeutic alternative for the treatment of aggressive pituitary tumors. Therefore, the paper is suitable for publication
However, there are some remarks:
- The results section should be modified, grouping paragraphs which share the same figure. In figure 1 and figure 2 (panels A and B) the authors should include the title of the Y axis.
- The discussion is a little confusing, in part because of the mixing of concepts of invasiveness and aggressiveness and in part because of the intention to explain an unexpected direct relationship between MSH6/12 and PD-L1 expressions. The concept of aggressiveness is a composite of invasion, proliferation (MIB-LI expression, but also others parameters such as strong nuclear p53 and high mitotic index), regrowth and recurrence. Indeed, the 2017 WHO classification of pituitary tumors do not longer recommend the term “atypical adenoma” based exclusively on the pathological parameters of proliferation. Because of the authors only take into account the MIB-LI index and moreover, the results are not conclusive, probably it would be better to omit this results and to focus on the invasiveness behaviour of the tumors studied.
- Material and Methods
3.1 Some mistakes: table 1, check 89 patients in the legend; the structure of the summarized data is confusing because, at first sight, it seems the data refers to the opposite subtypes instead of the whole series. In addition, it is more usual to use the term "radiological" instead of "imagological"
3.2 The authors should specify if the expression of MSH 6/2 and PD-L1 mRNA was calculated as number of copies or as fold change?
3.3 Why did the authors not measure the transcription factors in all the series instead of only in the tumors which were negative for pituitary hormones? The cut-off chosen (>1 %) to identify the pituitary tumor subtypes seems too low. Moreover, the percentage of NC tumors in the series seems too high. Both Nishioka (reference 6) and other authors (Torregrosa-Quesada ME, Cancer (Basel) 2019) reduced the percentage of NC tumors < 5 % in their respective series after studying the transcription factors, both by Immunochemistry and by RT-qPCR; most of the tumors being reclassified as GTs and also as CTs. The problem could be related with the SF-1 and Tpit antibodies used. Did the authors not find any tumor with unusual immohistochemical combinations? The precise identification of pituitary subtypes is essential to make comparisons between them. Therefore, the study of transcription factors should be performed in the whole series and the comparison between subtypes should be recalculated.
Minor aspects
- Page 3, line 93. Check 89 patients
- Page 7, lines 144 and 145. Add the dispersion of the percentages
- Page 9, lines 183. The reference Nº 20 should be substituted by the original one by Knosp
- Page 9, line 186. Delete "Highly proliferative"
- Page 9, line 192. The last sentence " The decrease in MSH 6/2.." should be rewritten
- Page 9, line 202. "However" is redundant with "contrary"
- Page 9, line 218. Last sentence " In NFPAs subtypes.." should also be revised
- Page 10, line 250. Substitute "clinically aggressive" by "clinically invasive"
Reviewer 2 Report
Comments
Results
Lines 144-149 Immunostaining was performed in 15 samples that are divided in 4 subgroups for the comparison. This number is probably too low to conclusive results. What are exactly the results of counting positive cells and heve been the results analyzed statistically. The proportions of the numbers of samples within these subgroups should be presented. Line 148-149 The statement - “Also, the expressions of PD-L1 mRNA was lower in NFPA subtypes with Knosp Grade 3-4 than in 148 NFPA subtypes with Knosp Grade 1-2 (Supplemental Figure 2).” – has it been statistically tested? P values are not presented in supplementary figure 2.
Materials and Methods
Lines 294 – 298 . Please provide the description of the method of calculating the expression levels. Description of y axis on figure 2 indicate that copy number of particular transcripts were calculated – how this was established?
Lines 323 -326 Description of statistical methods. What method of correlation testing and determining correlation coefficient was used? Has normal distribution been tested?. Supplementary figure 3 (, analysis MSH6 vs PDL1 (NCA)) suggest there are outlier samples in datasets normal distribution as the assumption for parametric test and Pearson correlation should be verified.
Lines 309-310 Description of western blot could be more detailed, at least the reference for the used protocol could be provided.
Line 252 The conlusion “Notably, decreased expressions of MSH6/2 may be useful for predicting the proliferation and invasiveness of NFPAs.” is too strong . MSH6/2 expression is not related to proliferation (MIB-1), according to the results of this study (supplementary figure 1)
Discusssion
I believe that when describing general role of MMR in tumors (lines 171 -173) the athor should mention that in human cancers MMR deficiency cause miscrosatelite instability and higher mutational load – a commonly accepted mechanism. In pituitary tumors low number of somatic point mutations are observed according to recent exome sequencing studies and MSI is not detected. Therefore if MMR proteins play a role in pathogenesis of pituitary it is not the same mechanism as in colorectal cancer (reference 24) .
Line 183 A simple classification was used of stratification into invasive and noninvasive gropus. It should be mentioned in the discussion that the classification may be more complicated and take also a visual inspection into account. Discussed in review (doi: 10.3390/cancers11121936.)
Line 179 The sentence “clinically NFPAs are often highly proliferative” is not exactly a true. NFPAs are generally benign and proliferation index is low as compared to other tumors.
195-196 “MIB-1 LI may not accurately represent pituitary tumor proliferation as the value is calculated in a hot spot of a pituitary tumor.” – this is not clear for me . What you mean “hot spot of a pituitary tumor”?
Line 223. I feel that the conclusion “tumor proliferation promotion due to decreased MSH6/2 expressions…,” is contrary to previous statement “The decrease in MSH6/2 expression may be due to the decrease in apoptosis rather than cell cycle promotion” (line 193)
254-256 “Considering MSH6/2 and PD-L1 expression patterns in NFPAs, ICIs may be effective against very aggressive NFPAs” - I don’t feel this conclusion is supported by the results. Agressive NFPAs were not included in the study.
Minor comments
- Please add the titles to axis on the graphs in supplementary file
- It is recently suggested to use the term pituitary tumor or PitNET instead of pituitary adenoma
- The results of diagnostic immunostaining could be added to supplementary table 1